# Effect of Polyaniline and Graphene Oxide Composite Powders on the Protective Performance of Epoxy Coatings on Magnesium Alloy Surfaces

Yingjun Zhang [1,2,*] , Shuai Xiao [1], Jie Wen [3] , Xinyu Liu [1], Baojie Dou [1] and Liu Yang [4]

1  College of Materials Science and Engineering, Sichuan University of Science and Engineering, Zigong 643000, China
2  Material Corrosion and Protection Key Laboratory of Sichuan Province, Sichuan University of Science and Engineering, Zigong 643000, China
3  Chengdu LEPS Technology Co., Ltd., Chengdu 610023, China
4  Chengdu Hongrun Paint Co., Ltd., Chengdu 610000, China
*  Correspondence: zhangyingjun@hrbeu.edu.cn

**Abstract:** Composite fillers are often used to improve the protective properties of coatings. To obtain a high protective performance of epoxy coatings for magnesium alloys, polyaniline (PANI) and graphene oxide (GO) composite powders were selected because of their corrosion inhibition and barrier performance, respectively. The paper mainly focuses on the effect of the preparation methods of the composite powders on the protective performance. PANI and GO composite powders were prepared by in situ polymerization and blending, respectively. First, the composite powder was characterized by X-ray diffractometer, Fourier transform infrared spectroscopy, and scanning electron microscopy. Then, the different composite powders and pure PANI powder were dispersed uniformly in epoxy resin, and the coating was prepared on the surface of the AZ91D magnesium alloy and studied by an electrochemical impedance test, adhesion strength test and physical properties test. The results show that the impedance value of the coating with the added PANI and GO composite powders by in situ polymerization was $4 \times 10^9 \ \Omega \cdot cm^2$ and higher than that with the added pure PANI ($4 \times 10^9 \ \Omega \cdot cm^2$) and PANI and GO mixed powders ($1 \times 10^9 \ \Omega \cdot cm^2$) after 2400 h immersion in a 3.5% NaCl solution; the former also had better flexibility, ss impact resistance, and adhesion strength. Compared with the direct blending method, the PANI and GO polymerization powders can exert the shielding effect of GO and PANI corrosion inhibition better and achieve a better protective effect on the magnesium alloy.

**Keywords:** magnesium alloy; polyaniline; graphene oxide; corrosion protection

## 1. Introduction

As the lightest metal material (65% of aluminum's density and 25% of iron's density) in engineering applications at present, magnesium alloys have the advantages of high specific strength, specific stiffness, and strong electromagnetic shielding ability. They also have broad application prospects in aerospace and other fields. However, their relatively high corrosion susceptibility and low potential (−2.37 V vs. SHE) limit their application in many transport applications [1–4]. Some surface treatments, such as anodizing, microarc oxidation, and chemical conversion film, are used to improve the corrosion resistance of magnesium alloys successfully [5–7]. However, the thickness or compactness is limited because the protection performance of the film is still dissatisfying.

Organic coating is a commonly used method in most metal protection methods because of its simple process, convenient construction, and excellent protection performance [8]. However, the long-term protection of magnesium alloys is difficult to achieve because of their high activity. One of the most common ways is to add different functional fillers to

increase the protective properties of the coating. The fillers can be divided into three main categories according to their mechanism of action. (1) Increased barrier performance of coating. Generally, some layered materials, such as montmorillonite [9], hydrotalcites [10], glass [11], and graphene [12], have attracted intense research interest because their lamellar elements increase the lengths of the diffusion pathways of oxygen, water, and aggressive ions [13]. In recent years, two-dimensional graphene oxide (GO) has attracted extensive attention because of its high specific surface area, nanosheet layer, and excellent barrier ability [14,15]. (2) Corrosion inhibition effect. The adding of some corrosion inhibition functional fillers into the coating is an approach. However, the high activity of magnesium alloys causes the selectivity of the fillers to be insufficient. Polyaniline (PANI) has been studied most widely because of its good stability, low cost, and unique doping mechanism [16–18]. In our previous works, the PANI coating achieved better protection against magnesium alloys because of its unique corrosion mechanism [19]. (3) Sacrificial anode protection. Some low-potential metals, such as Zn and Al, are used as sacrificial anodic protective fillers. However, it is difficult for the magnesium alloy matrix because of its lower potential.

In order to obtain the protective coating with an excellent comprehensive performance, composite fillers with the different functions mentioned above are added at the same time. In this paper, GO and PANI were added into the coating at the same time to improve the corrosion resistance for magnesium alloys by a combined effect. Some research found that the synthesized PANI/GO composite powders could improve the corrosion protection performance of coatings for steel [15,16,20], but there is not as much research on composite powder blending. Meanwhile, studies on magnesium alloys are limited. Therefore, this paper focuses on the effect of the coating prepared by added in situ polymerization and the direct blending compounding of PANI and GO on the protective properties of magnesium alloys.

## 2. Materials and Methods

### 2.1. Experimental Materials

Ammonium persulfate, aniline, sodium dodecylbenzene sulfonate, hydrochloric acid, and acetone were all analytically pure. The epoxy resin was E44 (purchased from Nantong Star Synthetic Materials Co., Ltd., Nantong, China), the curing agent was Cardolite LITE3100 (purchased from Caderai Chemical Co., Ltd., Zhuhai, China), and the commercial GO powder was produced by Suzhou Tanfeng Graphene Technology (Suzhou, China) Co., Ltd. The selected metal substrate was AZ91D magnesium alloy (with a chemical composition of Al 9.14%, Zn 0.86%, Mn 0.30%, Cu 0.09%, Si 0.08%, Fe 0.01%, and Ni 0.01%, and the rest was Mg, Dongguan Jiejin Metal Materials Co., Ltd, Dongguan, China). The magnesium alloy was surface-treated with 200 and 400 sandpapers in turn, washed with deionized water and acetone, and dried for later use.

### 2.2. Preparation of Composite Powder

A total of 0.1 g of sodium dodecylbenzene sulfonate was dissolved in 20 g deionized water in a three-necked flask by stirring. Then, we added 1 g of GO, dispersed at a high speed of 800 r/min for 1 h, named GO slurry. In addition, 10 g aniline was weighed into a three-necked flask, and hydrochloric acid was added dropwise at a speed of 400 r/min to control the pH within the range of 1 to 1.5 and stirred for 1 h. The dispersed GO slurry was added to a three-necked flask, and stirring continued for 1 h. After the stirring was completed, 24.5 g ammonium persulfate solution was added dropwise. After reacting for 12 h, the solution was washed with deionized water, filtered with suction until the filtrate was colorless, and dried to obtain PAGO. At the same time, pure PANI was prepared by the same synthesis process without GO, and PMGO also was prepared by blending PANI and GO directly according to the same ratio (PANI:GO = 10:1).

### 2.3. Preparation of Coating

The PANI, PAGO, and PMGO powders were added to the epoxy resin at 6% of the mass of the epoxy resin and dispersed at a high speed of 2000 r/min for 2 h. Then, the curing agent was added according to the mass ratio of 1:1.3 (epoxy: curing agent), stirred evenly, and then we coated it on the surface of the treated magnesium alloy. The cured thickness of the coating was (120 ± 15) µm, and a free film was prepared on the silica gel plate simultaneously.

### 2.4. Powder Characterization

The four kinds of powders were sputtered with gold before determined via scanning electron microscopy (SEM, VEGA3SBU, Tesken, Brno, The Czech Republic). Frontier near-infrared spectrometer (FT-IR, FT 9700, PerkinElmer, Waltham, Mass, USA) was used to identify whether the synthesized powder was indeed PANI and to find out the difference of the different compound powder at 400~4000 wavenumber region. The KBr pellet method was used to prepare FT-IR samples. X-ray Diffractometer (Bruker/D2 PHASER, XRD, Bruker, Karlsruhe, Germany) was conducted with a copper K$\alpha$ X-ray source.

### 2.5. Performance Test of Coating

Adhesion test was carried out using the BGD500 digital display pull-off adhesion tester produced by Biuged Laboratory Instruments (Guangzhou, China) Co., Ltd.

Autolab electrochemical workstation was used to test the performance of the coating in 3.5 wt.% NaCl solution. The 1 cm$^2$ platinum sheet and Ag/AgCl (saturated KCl, Huayu Instrument Co.,Ltd., Shanghai, China) were the counter and reference electrodes, respectively. The coated magnesium alloy sample was a working electrode with a test area of 9 cm$^2$. The test frequency was $10^{-2}$ to $10^5$ Hz, and the disturbance signal was a 30 mV sine wave.

According to the requirements of the ISO or Chinese standards (Table 1), the hardness, impact resistance, and flexibility of the coating were tested. All the test instruments were produced by Shanghai Modern Environmental Engineering Technology (Shanghai, China) Co., Ltd.

**Table 1.** Test standards and instructions of the physical properties of the coating.

| Test Property / Description | Standards | Instructions | Instruments |
|---|---|---|---|
| Film hardness | ISO 15184 [21] | Pencil test | PPH-1 pencil hardness tester |
| Flexibility | GB/T 1731-93 [22] | Bend test | QTX film flexibility tester |
| Impact resistance | GB/T 1732-93 [23] | Falling-weight test | QCJ impact tester |

## 3. Results

### 3.1. Analytical Characterization of Powder

Figure 1 shows the micromorphology of the synthesized PANI, GO, PAGO, and PMGO powders. PANI is an irregular globular and large agglomeration (Figure 1a). GO is distributed with an irregular flaky structure, and its surface is smooth. The accumulated lamellae may be due to the strong interaction between the surfactant groups after long-term storage (Figure 1b). Compared with GO, PAGO shows a smaller and thicker lamellar structure, and the GO sheet is surrounded by PANI particles (Figure 1c). The PANI particles are deposited uniformly on the GO surface. The particle size of PANI-deposited GO is smaller than that of pure PANI. With the adsorption and polymerization of aniline, GO was exfoliated into sheets and in steady state. Figure 1d shows the GO and PANI randomly scattered in the PMGO powder.

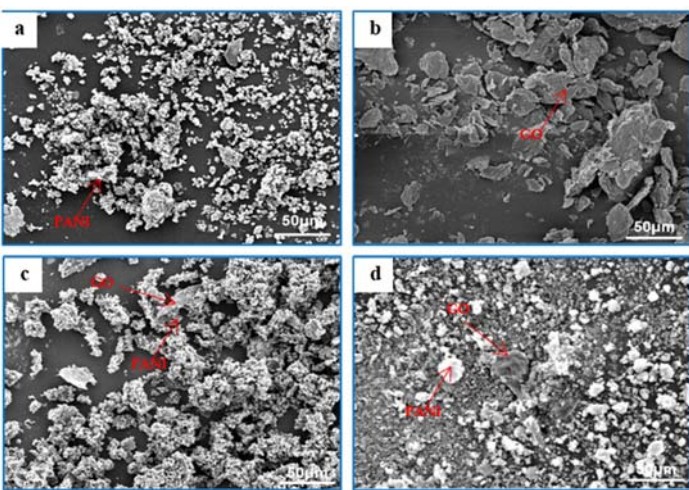

**Figure 1.** SEM of (**a**) PANI, (**b**) GO, (**c**) PAGO, and (**d**) PMGO powders.

Figure 2 shows the infrared test results of the PANI, GO, and PAGO powders. According to the literature [24–27], the characteristic absorption peaks that appear at 1572 and 1465 cm$^{-1}$ in the infrared spectrum of PANI are the C=C bending vibration of the quinone ring and the C=C bending vibration of the benzene ring, respectively. The peaks at 1298 cm$^{-1}$ correspond to the C–N stretching of a secondary aromatic amine; the peaks at 1104 cm$^{-1}$ are assigned to vibrations associated with the C–H of the quinone ring. For GO, the characteristic absorption peaks at 1043, 1224, and 1718 cm$^{-1}$ are the C–O–C stretching vibration of the GO surface, the C–O stretching vibration of the carboxyl group, and the C=O stretching vibration of the carboxylic acid, respectively [28]. The characteristic absorption peaks of the PMGO powders included PANI and GO and did not significantly change, which indicated that there was no reaction between them. However, compared with PANI and GO, all the characteristic absorption peaks of PANI appeared in the infrared spectrum of the synthesized PAGO powder, but some characteristic absorption peaks of GO, such as the C=O stretching vibration of the carboxylic acid (1718 cm$^{-1}$), disappeared owing to a partial peak overlap and coverage, indicating that PANI was successfully polymerized on the GO surface.

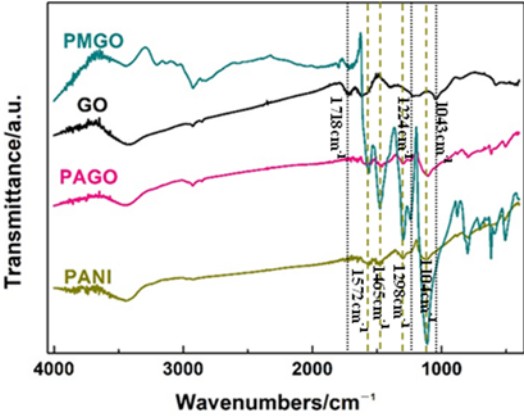

**Figure 2.** Infrared spectrum of synthetic powder.

Figure 3 shows the XRD test results of the PANI, GO, PAGO, and PMGO powders. The PANI synthesized in the figure has characteristic diffraction peaks at 2θ of 20.3° and 25.0°, and its diffraction peaks are relatively broad, indicating that the state is partially crystalline [29]. GO has a strong diffraction peak at 2θ of 11.0°, indicating an increased degree of oxidation and disorder of the graphite sheet [30]. The characteristic diffraction peaks that appeared in the synthesized PAGO are consistent with those of PANI, and the

diffraction peak that corresponded to GO almost disappears, suggesting that GO is completely surrounded by PANI. However, the XRD pattern of the PMGO powder contained the diffraction peaks of PANI and GO, indicating the separate situation between the PANI and GO powders. But compare to the strong diffraction peak of GO, the diffraction peak of PANI is inconspicuous. In addition, these results confirm that PANI was successfully polymerized on the GO surface.

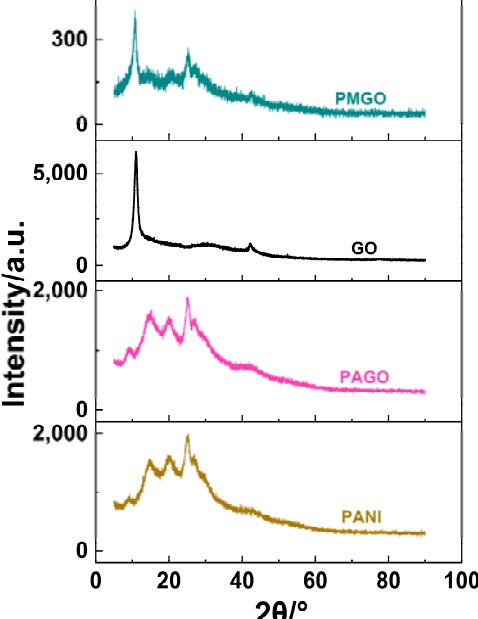

**Figure 3.** XRD of the synthesized powders.

### 3.2. Coating Performance Analysis

### 3.2.1. Analysis of the Physical Properties of Coatings

Table 2 shows the test results of the physical properties of the PANI, PAGO, and PMGO coatings. Compared with the PANI coating, the hardness, flexibility, and impact resistance of the PMGO coating are improved when GO is added through blending, whereas the impact resistance is further improved and the flexibility is also enhanced for the PAGO coating when GO is introduced by polymerization, because the GO lamellar structure can disperse the stress applied on the coating. When PANI is blended with GO, granular PANI and lamellar GO are randomly distributed in the coating. When PANI and GO are polymerized in situ, PANI is uniformly distributed in the coating after PANI is deposited on the GO surface (Figure 1c). Thus, it can play a more effective role in toughening graphene.

**Table 2.** Test results of the physical properties of the coating.

| Property Coating | Hardness | Flexibility | Impact Resistance (1 kg) |
|---|---|---|---|
| PANI | 3H | Diameter Φ4 mm | 24 cm |
| PAGO | 4H | Radius of curvature 0.5 ± 0.1 mm | 35 cm |
| PMGO | 4H | Radius of curvature 1.5 ± 0.1 mm | 28 cm |

Figure 4 shows the water absorption (marked as *Q*) of PANI, PAGO, and PMGO free film. *Q* can be determined using the following formula.

$$Q = \frac{w_t - w_0}{w_0},$$

where $w_t$ (g) is the amount of absorbed water at $t$ (s) time, and $w_0$ (g) is the initial weight before immersion. Changes in $Q$ of different coatings are roughly similar in Figure 4. That is, they all rise rapidly at the beginning and reach a stable state with the increase in time, and the saturated water absorption ($Q$-saturation) of the three kinds of coatings are all maintained at a low level, of which the Q-saturation of the PANI coating is the highest at 1.48%, whereas that of the PAGO coating is the lowest at 1.12%. The GO lamellar structure can disperse the shrinkage stress during the curing period, which is helpful for forming a denser coating. Therefore, the defects of the coating are reduced, and the $Q$-saturation of the coating is low.

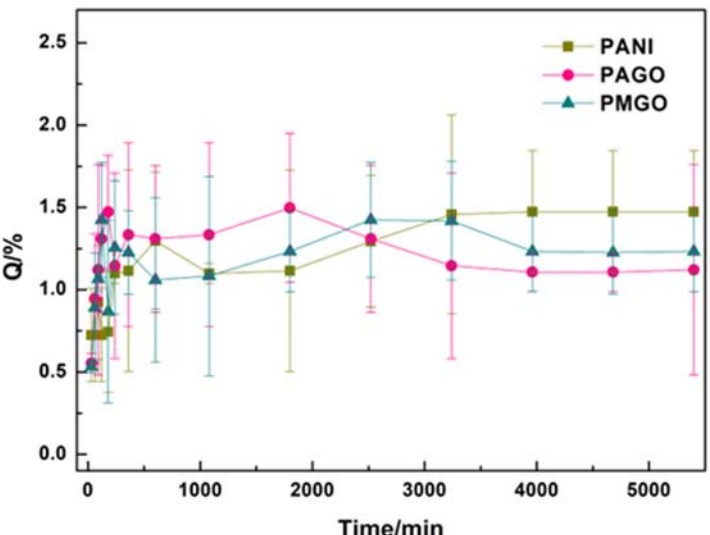

**Figure 4.** Variation curve of water absorption rate of different coatings.

3.2.2. Analysis of Adhesion Strength of Coatings

Figure 5 shows the adhesion strength value and surface morphologies of the dry and wet adhesions of different coatings (gray column chart shows the test results of the wet adhesion strength of different coatings after 1200 h immersion). The dry adhesion strength of the PANI, PAGO, and PMGO coatings are 9.6, 11.5, and 10.7 MPa, respectively. After the test, only part of the metal matrix observed all three kinds of coatings, which also indicates high adhesion strength. Compared with the pure PANI coating, the adhesion strength of the PAGO and PMGO coatings was enhanced by the addition of GO. The main reason may be that the lamellar structure of GO can reduce the stress concentration of the coating. The adhesion strength of the PAGO coating is higher than that of the PMGO coating because GO was dispersed better by in situ polymerization than by direct mixing, and GO can play a better role.

Meanwhile, the PANI, PMGO, and PAGO coatings could also be defined as composite material, epoxy resin as the matrix (continuous phase), and different kinds of fillers as the reinforced (dispersed) phase. Therefore, the fracture of the coating during the adhesion test could refer to the idea of the mechanics of composite materials. Some novel and strong models, such as the "Tsai-Wu"and "Checkerboard" models, have been recently proposed to estimate the strength of epoxy-reinforced glass or graphene specimens. According to the literature, a very little volume of graphene nanoplatelets would double the critical buckling load of the transverse-oriented fiber composite by calculation, which can be attributed to the significant increment of the matrix modulus of elasticity [31,32].

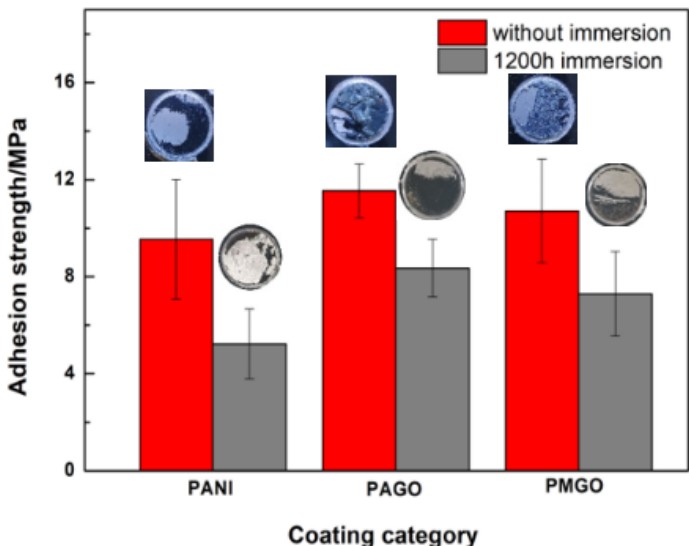

**Figure 5.** Dry and wet adhesions of the three kinds of coatings.

The wet adhesion strength of the PANI coating is 5.2 MPa, and almost all of the coatings are stripped, and some gray product film is formed. The wet adhesion of the PAGO coating is 8.4 MPa, and only a small portion of the coating was removed. The wet adhesion of the PMGO coating is 7.3 MPa, and about half of the coating was stripped. The wet adhesion strength of the PMGO and PAGO coatings is higher than that of the PANI coating, because the labyrinth effect caused by the GO lamellar structure delays the infiltration of the corrosion medium. Thus, the adhesion strength is enhanced. Compared with the PMGO coating, the wet adhesion strength is higher, and the stripping area of the PAGO coating is smaller because the dispersion effect of GO by in situ polymerization is better than that by direct mixing, and the shielding effect is stronger.

3.2.3. Analysis of Coating Protection Performance

Figure 6 presents the Nyquist and Bode diagrams of the EIS of the PANI, PAGO, and PMGO coatings in different time periods. The low-frequency impedance modulus of the coating can be used to characterize the protective performance of the coating. In this study, the impedance value of the coating was taken when the frequency was 0.01 Hz. Figure 7 shows that the three kinds of coatings all show a rapid decrease at first and then a relatively stable state. The reason is that at the initial stage of immersion, with the increase in time and the penetration of the solution, the shielding property of the coating decreases rapidly when the water absorption reaches saturation. The performance of the coating tends to be stable. The high modulus of the PAGO coating in the early stage of immersion is due to the lamellar shielding effect of GO. In addition, in the process of soaking for 2400 h, the values of the three kinds of coatings are all higher than $10^8 \Omega \cdot cm^2$, indicating that the coatings have a protective effect on the magnesium alloy, and the values of the coatings with the PAGO powder coating are higher than those of the others, indicating that the coatings have a better protective performance.

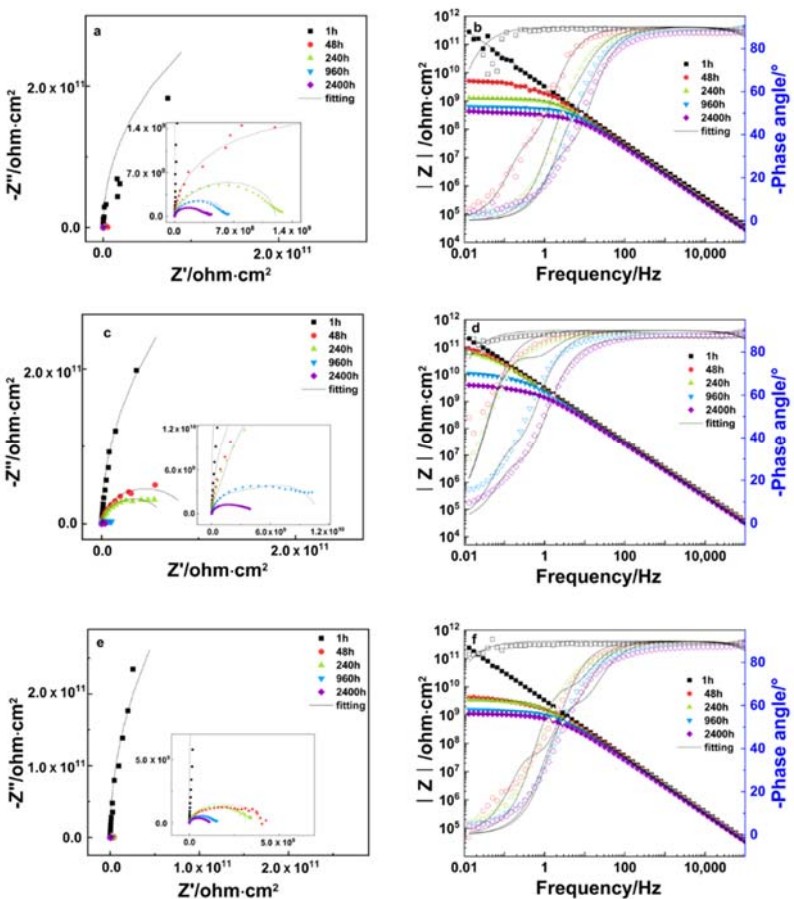

**Figure 6.** EIS spectra of three kinds of coatings. (**a,b**)—PANI; (**c,d**)—PAGO; and (**e,f**)—PMGO.

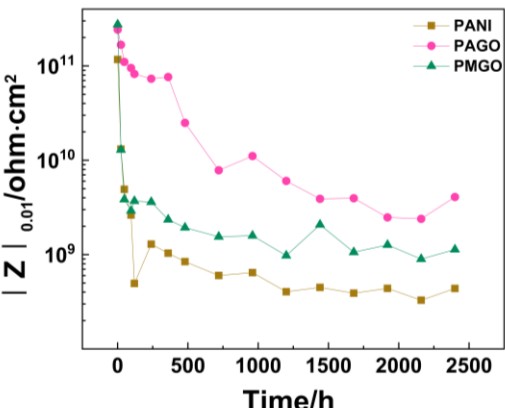

**Figure 7.** Low-frequency impedance value of the three coatings versus time.

The equivalent circuit diagram of Figure 8 was selected to fit the data. Figure 8a was selected for data fitting during the immersion process, when there is a time constant in the impedance spectrum of the coating; that is, there is only one capacitive arc in the Nyquist diagram, and there is no platform in the Bode curve in the low-frequency region. This indicates that the corrosive species penetrated into the coating but did not reach the coating/substrate interface. Meanwhile, Figure 8b was selected for data fitting when the platform appears in the low-frequency region, suggesting that the corrosive agent has penetrated the coating and reached the coating/substrate interface. $R_s$ represents the solution resistance; $Q_c$ represents the coating capacitance; $R_{coating}$ represents the coating resistance; $Q_{dl}$ represents the electric double layer capacitance at the coating–metal interface,

and $R_t$ represents the charge transfer resistance. The red solid line in Figure 6 is the result of data fitting.

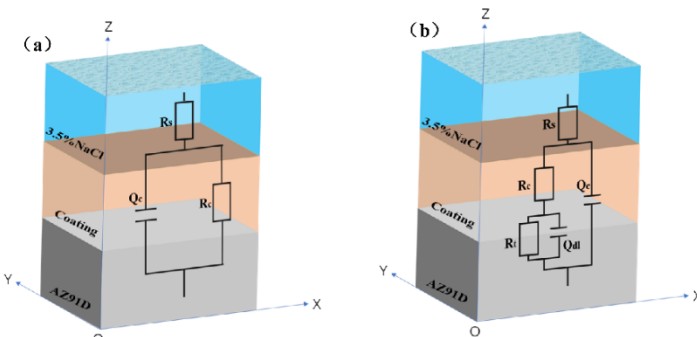

**Figure 8.** Equivalent circuits of coating/magnesium alloy systems (Models A and B).

### 3.2.4. Analysis of Coating Protection Mechanism

(1) Shielding Effect of the Coating

Figure 9 shows the variation curve of the coating resistance ($R_{coating}$ of the PANI coating, PAGO coating, and PMGO coating during 2400 h immersion after equivalent circuit fitting. $R_{coating}$ reflects the barrier property of the coating [33], which is an important factor for characterizing the protective performance of the coating. The $R_{coating}$ values of the three coatings decreased rapidly at first and then tended to be stable with the increase in immersion time. The rapid decrease in the early stage was mainly caused by water absorption in the process of coating soaking, and the later stage tended to be stable because the water absorption reached the saturation state. The PAGO coating and the PMGO coating were higher than the PANI coating because the GO coating had better shielding performance, whereas the PAGO and PMGO coatings had a larger difference because of their different dispersion degrees in the coating, which affected the compactness of the coating. At the same time, the compatibility between the powder and coating in the PMGO coating was poor, thereby resulting in the lower $R_{coating}$ value of the PMGO coating than that of the PAGO coating. Such a result is consistent with the Q-saturation of the coating (Figure 4). The results show that the coating with PAGO powder had better resistance to solution penetration. Thus, the PAGO coating had a better protective effect on the AZ91D magnesium alloy than other coatings.

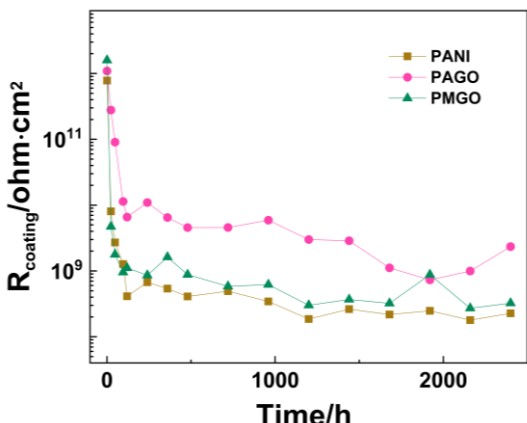

**Figure 9.** Variation curve of coating resistance with immersion time.

(2) Corrosion inhibition effect of coating

Figure 10 shows the variation curve of the $R_t$ values of the PANI, PAGO, and PMGO coatings during 2400 h immersion. The higher the $R_t$ value is, the smaller the corrosion rate of the metal substrate will be [34]. Therefore, $R_t$ is inversely proportional to the corrosion

rate of the metal. During the whole immersion process of the coating, the $R_t$ values of the PANI, PAGO, and PMGO coatings showed a fluctuation of decrease-increase-decrease, which is the result of the joint action of corrosion of the magnesium alloy and the corrosion inhibition of PANI. Among them, the $R_t$ value of PAGO and PMGO coating is higher than that of the PANI coating, which is mainly due to the shielding effect of GO, which ensures that PANI can inhibit the corrosion of the magnesium alloy effectively. The $R_t$ value of the PAGO coating is higher than that of the PMGO coating because the coating has better shielding properties (Figure 9). Therefore, compared with the other coatings, the addition of the PAGO powder coating has a better protective effect on the AZ91D magnesium alloy.

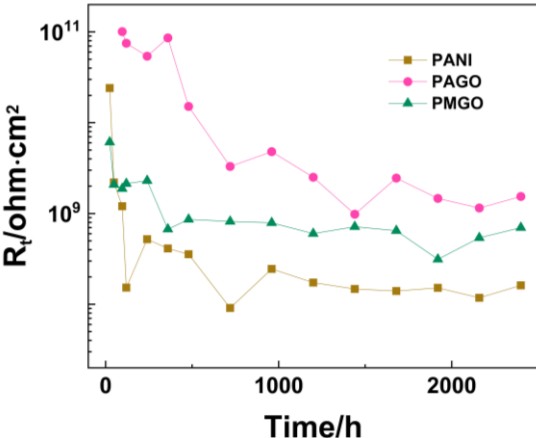

**Figure 10.** Change curve of charge transfer resistance with immersion time.

(3) Protection mechanism of coatings

Figure 11 shows the protection mechanism of different coatings. For the PANI coating, some defects were formed during hardening because of the solvent's volatility, PANI particle agglomeration, and shrinkage stress. These defects provide the initial channel for solution penetration. With the increase in soaking time, some new diffusion channels formed because of water polarization and osmosis. When the aqueous solution reached the magnesium alloy surface, the oxidation-reduction action of PANI formed a protective product film [19]. For the PAGO coating, shrinkage stress would be reduced because of the excellent flexibility of the GO sheets, and the PAGO particles would be dispersed more uniformly, thereby making coatings with fewer defects on the diffusion channel. Therefore, the PAGO coating had an excellent shielding performance. Similarly, PANI formed a protective product film when the aqueous solution reached the magnesium alloy surface. For the PAGO coating, shrinkage stress would be reduced because of the excellent flexibility of the GO sheets, and the PAGO particles would be more uniformly dispersed, therefore making a coating with fewer defects on the diffusion channel. Therefore, the PAGO coating had an excellent shielding performance. Similarly, PANI formed a protective product film when the aqueous solution reached the magnesium alloy surface. For the PMGO coating, the PANI particles and GO sheets dispersed unevenly in the coating, which would influence the shielding performance of the coating, though shrinkage stress would be reduced because of the excellent flexibility of the GO sheets.

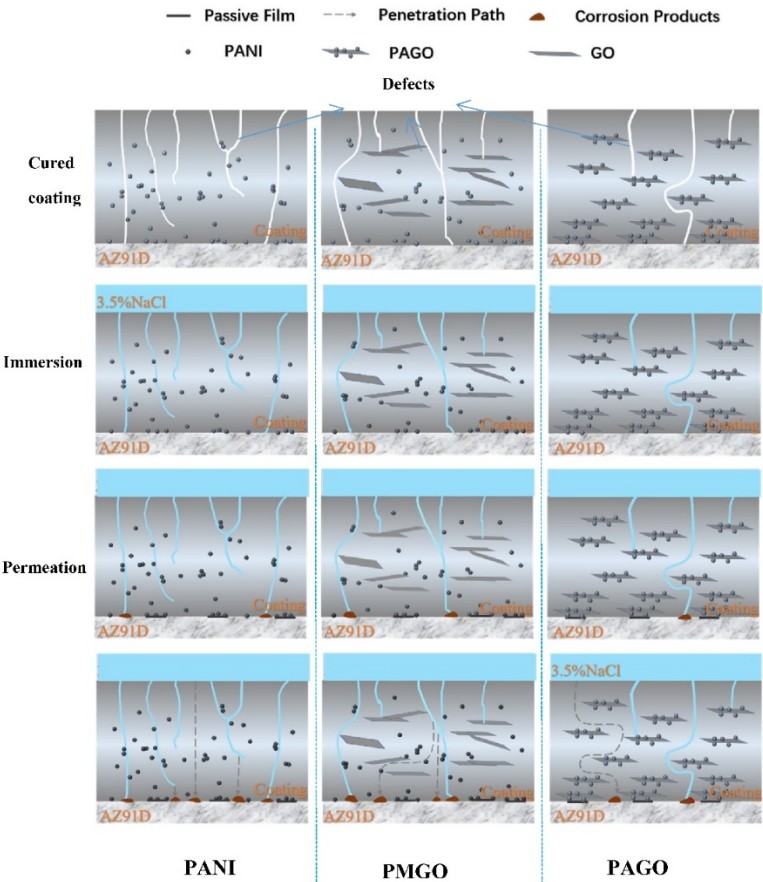

**Figure 11.** Schematic of the protection mechanism of different coatings.

## 4. Conclusions

In this study, the physical and corrosion protective performances of epoxy coatings containing three kinds of fillers (PANI, PMGO, and PAGO) were compared. GO/PANI were prepared and applied on mild steel. The results of various tests show that (1) the composite powder containing GO could improve the performance of the PANI coating whether it was prepared by blending or the polymerization method; (2) compared with the blending method, the composite powder prepared by the polymerization method had a better physical performance and corrosion protective effect on the magnesium alloy; (3) the PAGO coating had a better shielding performance because of its fewer defects, and it was more uniformly dispersed. Similarly, PANI formed a protective product film when the aqueous solution reached the magnesium alloy surface.

The EIS results indicate that the PAGO coating had outstanding corrosion protective performance for the magnesium alloy after 2400 h. This was an exciting result because it is difficult to protect magnesium alloys from corrosion. However, the influence of the type and ratio of PANI and GO on the properties of the coatings will be discussed further.

**Author Contributions:** Y.Z.: Conceptualization, Writing—original draft, Writing—review and editing, and Supervision. S.X.: Resources and Investigation. J.W.: Methodology. X.L.: Resources. B.D.: Data curation and Writing—review and editing. L.Y.: Investigation. All authors have read and agreed to the published version of the manuscript.

**Funding:** This work was supported by the National Natural Science Foundation of China (No. U21A2045 and No.51801131), Sichuan Science and Technology Program (No.2022NSFSC0300 and No.2022ZHCG0076).

**Institutional Review Board Statement:** Not applicable.

**Informed Consent Statement:** Not applicable.

**Data Availability Statement:** Not applicable.

**Conflicts of Interest:** The authors declare no conflict of interest.

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
