# Peer review of "Effect of Polyaniline and Graphene Oxide Composite Powders on the Protective Performance of Epoxy Coatings on Magnesium Alloy Surfaces"

_coatings, doi:10.3390/coatings12121849_

Round 1

Reviewer 1 Report

Please read and “fully” address the comments listed below: 

1.     The ABSTRACT is not written in a logical order. Start with an overview of the topic and a rationale for your paper. Describe the methodology you used and the general outline of the manuscript. Also, in the end, state the result in more detail (i.e., provide some numbers).

2.    The novelty of your work is still unclear to the reader, which should be further detailed both in the Abstract and Introduction.

3.     Please add an error bar to the bar charts where needed, e.g., Fig. 4.

4.     In Fig11 (or similar plots), please change the color map to a perceptually uniform (color-blind friendly) scale. 

5.    Please fully introduce the elastic properties of all the structural components explained in the manuscript. You can summarize them in a table.

6.    Include XYZ coordinates to all figures, where needed, e.g., in Fig 8.

7.    Please fully explain the process used for mixing and casting the composite specimens.

8.    Page 2, line 52, please rewrite this sentence “However, the high activity of magnesium alloy causes the selectivity of fillers to be insufficient.”

9.     In Fig. 1, label each component of PANI, GO, PAGO, and PMGO powders images.  

10.  Apart from experimental results, more novel and strong models have been recently proposed to find the shear strength of composite specimens. Among them, the “Tsai-Wu” and “Checkerboard” models proved that they could estimate the strength of epoxy-reinforced graphene specimens. For this purpose, please write a paragraph in your paper introducing these models which can “alternatively” provide the adhesion strength of your composites, and reference the manuscript listed below

Tsai-Wu

Chen, X., Sun, X., Chen, P., Wang, B., Gu, J., Wang, W., ... & Zhao, Y. (2021). Rationalized improvement of Tsai–Wu failure criterion considering different failure modes of composite materials. Composite Structures, 256, 113120.

Checkerboard:

Kabir, H., & Aghdam, M. M. (2019). A robust Bézier based solution for nonlinear vibration and post-buckling of random checkerboard graphene nano-platelets reinforced composite beams. Composite Structures, 212, 184-198. 

11. Conclusion: Can authors highlight future research directions and recommendations? Also, highlight the present study’s assumptions and limitations (shortcomings)). Besides, recheck your manuscript and polish it for grammatical mistakes (you can use “Grammarly” or similar software to quickly edit your document).

Reviewer 2 Report

The manuscript entitled "Effect of polyaniline and graphene oxide composite powder on 2 the protective performance of epoxy coating on magnesium al- 3 loy surface"- An interesting knowledge has been propose but the following comments should be addressed before further progress

1) introduction part can be enriched by providing some Qualitative data

2) the novelty of the manuscript must be better emphasized in introduction part

3) Suggested to compare some other materials also with  magnesium alloy for its better properties and unique characters 

4) 2.3. Powder characterization part should mentioned elaborately

5) Frontier near-infrared spectrometer and potassium bromide 97 tablet method was applied to analyze the synthesized powder - sentence not properly phrase

6) in figure 2 suggested to mention the wavenumbers inside to the figure, that could be easy to understand 

7) page number 3, IR interpretation can be more discussed

8) in overall characterization studies, suggested to add some more discussion points

9) Suggested to add a  comparison table for physical properties of the coating with previously published reports

10) suggested to add more clear conclusion statement 

11) some of the references are too old, suggested to update

After addressing all the comments, this manuscript can be proceed for further progress 

Reviewer 3 Report

The Manuscript “Effect of polyaniline and graphene oxide composite powder on the protective performance of epoxy coating on magnesium alloy surface” by Yingjun Zhang et al. devoted to polyaniline and graphene oxide composite synthesis for corrosion inhibition process on the surface of the AZ91D magnesium alloy. Corrosion protection by applying protective coatings to the surface of working metals is an actual subject for both scientific research and industrial applications. Therefore, the Manuscript deserves publication in the Coatings journal.

There are a few minor questions and comments 

1.     Materials and methods. Some of elements were written as symbols (for examples, Al, Mn etc.), but some of them – as full name (copper and zinc). It is necessarily to used one style of naming

2.     It would be better to indicate more precisely the parameters of the ongoing XRD and IR spectroscopy studies (step, speed, etc.) as well as the parameters of the SEM

3.     Why was a surfactant introduced into a system with graphene oxide? Is it possible that the presence of surfactants gives some additional properties to the system and improves anticorrosion properties or not?

4.     The strength properties of graphene oxide are not equivalent to ones of graphene. In this case, everything is determined by the polymer film coating of the metal layer: polymer composite also can percolate through the nano and micropores inside the alloy. This contributes to additional adhesion strengthening. The graphene oxide powder is also small enough for uniform permeation, and the undivided GO layers provide an impediment (sliding of the layers relative to each other).

5.     Surprisingly, when GO is added to the coating, its Q saturation (water sorption) decreases, although graphene oxide is a rather hydrophilic material. Apparently, this really indicates to complete closure of GO sheets with polyaniline.

Round 2

Reviewer 1 Report

The authors addressed my comments and the manuscript can be published in the present format.